# A Two-Stage Structural Damage Detection Method Based on 1D-CNN and SVM

Chenhui Jiang [1] , Qifeng Zhou [1,*] , Jiayan Lei [2] and Xinhong Wang [2]

1   Department of Automation, Xiamen University, Xiamen 361005, China
2   Department of Civil Engineering, Xiamen University, Xiamen 361005, China
*   Correspondence: zhouqf@xmu.edu.cn

**Abstract:** Deep learning has been applied to structural damage detection and achieved great success in recent years, such as the popular structural damage detection methods based on structural vibration response and convolutional neural networks (CNN). However, due to the limited number of vibration response samples that can be acquired in practice for damage detection, the CNN-based models may not be fully trained; thus, their performance for identifying different damage severity as well as the damage locations may be reduced. To solve this issue, in this paper, we follow the strategy of "divide-and-conquer" and propose a two-stage structural damage detection method. Specifically, in the first stage, a 1D-CNN model is constructed to extract the damage features automatically and identify the damage locations. In the second stage, a support vector machine (SVM) model and wavelet packet decomposition technique are combined to further quantify the damage. Experiments are conducted on an eight-level steel frame structure, and the accuracy of the experimental results is greater than 99%, which demonstrates the superiority of the proposed method compared to the state-of-the-art approaches.

**Keywords:** structural damage detection; convolutional neural networks; support vector machine; multi-level damage classification

## 1. Introduction

Structural damage is inevitable and more likely to happen when a variety of mechanical or environmental elements are present. Structural damage can decrease the life of a structure and threaten people's safety. Establishing a structural health monitoring (SHM) system , which is also crucial for enhancing structural reliability and safety and lowering maintenance costs, is an efficient solution to solve this issue [1]. SHM is a multidisciplinary research field that includes experimental testing, system identification, data collecting and management, and long-term environmental data measurement [2,3]. The most important part of SHM is structural damage detection (SDD), which is a methodical, automated process for detecting damage, locating it, and determining its severity [4].

SDD begins with visual inspection, but visual inspection has numerous limitations. First, due to the generally high scale of civil construction, routine inspections are time-consuming and laborious. Second, the inspector's specific knowledge is required for visual inspection. Third, load-bearing structures are frequently hidden behind flooring, ceilings, and other decorative materials, making visual inspection impossible [5]. With the continued advancement in the fields of structural health monitoring and structural damage detection, numerous strategies have been employed to identify, locate, and quantify structural damage to overcome the limitations of visual inspection [6–8].

Damage to a structure alters the structure's mass and stiffness distribution, causing changes in the inherent frequency and vibration pattern [9]. Many approaches based on structural vibration response , which extract features from the vibration response and then determine the corresponding damage state, have been presented [10]. Some traditional

machine learning algorithms have been applied to this field, with support vector machine (SVM) being one of the most classic. Lei et al. [11] proposed a method based on vibration statistical indicators and SVM, employing variance, regression coefficients, and cross-correlation function amplitude as feature vectors for SVM, and achieved good results on an eight-level steel frame structure. An enhanced Hilbert–Huang transform (HHT) and SVM-based structural damage detection technique is proposed by Diao et al. [12]. The structural vibration function's Hilbert spectral energy is obtained by decomposing the vibration signal using improved empirical modal decomposition. Then, the structural damage feature vector is constructed, and the damage location and severity are detected using SVM. The method's efficacy is evaluated on an experimental model of an offshore platform.

However, feature extractors based on traditional machine learning techniques require significant domain expertise, whereas deep-learning-based methods can automatically extract and select features from data, thus avoiding the need for the manual design of feature extraction methods and reducing workload [13]. In the field of structural health monitoring based on vibration response, one-dimensional convolutional neural networks (1D-CNN) are very popular deep learning methods; 1D-CNN takes time series directly as input and conducts one-dimensional convolution on the time axis. Ma et al. [14] used 1D-CNN to detect damage in a steel beam numerical model. According to their results, 1D-CNN based on acceleration signals could detect 94.1% of the damage. Wang et al. [15] used the time-frequency graph of the damaged signal after HHT transform and the marginal spectrum of the signal as the input of CNN and optimized the parameters of CNN with particle swarm optimization (PSO) to improve the model performance. Xiao et al. [16] proposed an improved denoising auto-encoder-based neural network and optimized it by using gray relation analysis. It is capable of automatically extracting high-level features from the original signal by multi-layer extraction and can achieve high accuracy in noisy environments. In addition, numerous works have demonstrated that 1D-CNN outperforms traditional machine learning methods in structure damage detection.

In applications of SHM, different users have different requirements. For example, a house owner only needs to know the location of damage to his house to contact the maintenance staff, while the maintenance staff needs to know more details about the damage, including the location and severity of the damage to make better repairs. Therefore, it makes sense to design a multi-stage structural damage detection method to save computational costs.

To enhance both the model performance for detecting the damage location and the damage severity, in this paper, we propose a two- stage structural damage detection method in which the selection of a classifier for each stage is very important. We choose 1D-CNN to detect the damage location because 1D-CNN has been widely adopted in the field of SDD with good results. It was also discovered in [17] that 1D-CNN is more accurate for identifying the damage location than the damage severity. This is because the variety of structural vibration responses on different damage locations is much larger than that of the damage severity. In addition, there is a very small difference between the vibration response corresponding to different damage severity at the same location. In this case, a CNN-based model may need more complex structures and more training data. Therefore, after detecting the damage location, we can choose another method that may produce better results with fewer samples and classes to detect the damage severity instead of detecting the damage location and severity at the same time.

SVM is a strong classification machine for small-scale sample learning problems. It is a sparse kernel decision machine that avoids computing posterior probabilities when building its learning model. SVM has been extensively used for classification, regression, novelty detection tasks, and feature reduction [18]. Compared with 1D-CNN, SVM is more suitable for damage severity identification, and the computational cost and required training samples of SVM are less than those of 1D-CNN.

In summary, the novelty and the main contributions of this paper are as follows:

- We propose a new two-stage structural damage detection method which follows the strategy of "divide-and-conquer" to solve the problem of insufficient training data and enhance the model performance for multi-level structural damage detection.
- Our method fully combines the advantages of 1D-CNN and SVM, reducing computational costs and eliminating the need to rely on expertise to design complex feature extraction methods.
- We verify the proposed model on an eight-level steel frame structure. The experimental results show that the proposed method outperforms the state-of-the-art methods in terms of both damage location detection and damage severity detection.

## 2. Methods

The framework of the proposed two-stage structural damage detection method based on 1D-CNN and SVM is shown in Figure 1. After data preprocessing, the samples are identified using a two-stage approach. In the first stage, the samples are classified according to the damage location using 1D-CNN. In the second stage, the frequency domain features of the samples are extracted using wavelet packet decomposition. Then, the feature vectors are learned using support vector machine, and the damage severity of the samples can be obtained.

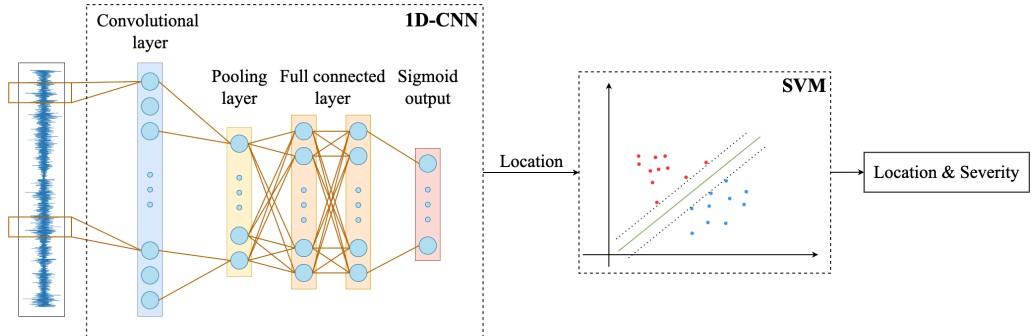

**Figure 1.** An overview of our proposed framework.

### 2.1. 1D-CNN

In this work, 1D-CNN is adopted as the classifier for the first stage. Using 1D-CNN to extract features from time series is a natural way. Compared with the traditional approaches, 1D-CNN directly takes a one-dimension time series as input, without the need to understand the physical meaning contained in the time series. Therefore, we can construct a 1D-CNN model to automatically extract rich features from structural vibration response and then classify the structural damage locations. The basic 1D-CNN model includes three parts: convolutional layer, pooling layer, and fully connected layer. The 1D-CNN in practical applications contains more components to improve the model's performance. The 1D-CNN model constructed in this work includes convolutional layers, pooling layers, global average pooling layers, dropout layers, fully connected layers, and softmax output layers.

#### 2.1.1. Convolutional Layer

The convolutional layer uses a time window sliding along the time axis direction of the time series to obtain a set of subsequences and then multiplies each subsequence with the kernel element-by-element to obtain the convolution result, as shown in Equation (1). The convolutional layer has three characteristics: sparse weights, parameter sharing, and equal variation [19]. These properties significantly reduce the model's memory cost and improve the model's ability to extract data features automatically.

$$y(k) = \sum_{i=0}^{N} h(k-i)u(i), \tag{1}$$

where $h$ represents the subsequence, $u$ (i) represents the kernel, $y$ represents the output signal, $k$ represents the index of the subsequence, and $N$ represents the length of the kernel.

ReLU [20] is chosen as the activation function in the convolutional layer, which has less computational overhead and faster computation than activation functions such as Sigmoid. The formula of ReLU is as in Equation (2). When the input $x$ of ReLU is non-negative, the output result is $x$. When $x$ is less than 0, the output result is 0.

$$ReLU(x) = \begin{cases} x, & x \geq 0 \\ 0, & x < 0 \end{cases} \tag{2}$$

### 2.1.2. Pooling Layer

The pooling layer can reduce the output feature dimension of the convolutional layer by downsampling [21]. In this paper, the max pooling method is used to reduce the feature dimension, which takes the maximum value in the neighborhood as the representation of the neighborhood. In this paper, we utilize global average pooling to compress the high-dimensional feature vector output from the convolutional layer to one dimension vector as the input of the subsequent model. Global average pooling aggregates the feature information of each dimension, which is more robust to the noise in the feature vector [22].

### 2.1.3. Droput Layer

Dropout is an effective tool for solving the overfitting problem [23]. Dropout operation is applied to the output of the global average pooling layer. Dropout is based on randomly masking some units during training and enabling them during validation, which can effectively improve the performance of CNN.

### 2.1.4. Full Connected Layer

After several layers of convolution and pooling, all information needs to be integrated from the hidden feature space using the fully connected layer to complete the damage detecting task.

### 2.2. SVM

SVM is a very effective machine learning technique widely used in classification, regression, anomaly detection, and other learning tasks [24,25]. Given the training samples and labels $x_i \in \mathbb{R}^n, y_i \in \{-1, +1\}, i = 1, \ldots, m$, SVM can solve the following optimization problem:

$$\begin{aligned} &\min_{w,b,\xi_i} \frac{1}{2}\|w\|^2 + C\sum_{i=1}^{m} \xi_i \\ &\text{s.t. } y_i\left(w^{\mathrm{T}}x_i + b\right) \geqslant 1 - \xi_i, \\ &\quad \xi_i \geqslant 0, i = 1, 2, \ldots, m, \end{aligned} \tag{3}$$

where $w$ represents the weight and $y$ is the sample label, $\xi_i$ is the relaxation variable, C is the penalty coefficient, m is the number of samples, and b is the bias. SVM finds a linear partitioned hyperplane with maximum margin in a high-dimensional space. By solving this optimization problem for $w$, $b$, and $\xi_i$, the optimal hyperplane which can be used to classify samples is obtained. This optimization problem can be solved with the help of the Lagrange multiplier method or quadratic programming. However, this optimization

problem is usually complicated and needs to be solved with the help of the dual problem. The dual form is given by Equation (4).

$$
\max_{\boldsymbol{\alpha}} \sum_{i=1}^{m} \alpha_i - \frac{1}{2} \sum_{i=1}^{m} \sum_{j=1}^{m} \alpha_i \alpha_j y_i y_j \boldsymbol{x_i}^T \boldsymbol{x_j}
$$
$$
s.t. \sum_{i=1}^{m} \alpha_i y_i = 0,
$$
$$
0 \leq \alpha_i \leq C \quad i = 1, 2, \ldots, m,
$$
(4)

where $\alpha_i$ represents the Lagrange multiplier. After solving the dual problem, the solution of the original problem can be obtained:

$$
\boldsymbol{w} = \sum_{i=1}^{m} \alpha_i y_i x_i,
$$
(5)

$$
b = y_j - \sum_{i=1}^{m} \alpha_i y_i x_i^T x_j
$$
(6)

SVM can be extended to a nonlinear classifier by introducing kernel functions. There are many commonly used kernel functions such as linear, polynomial, sigmoid, and radial basis function (RBF) [26]. Among them, the RBF function is the most widely used kernel function [27] because it has a solid ability to distinguish the non-linearly separable data. The formula of the RBF function is shown in Equation (7).

$$
\mathbf{K}(x_i, x_j) = \exp\left(-\frac{\|x_i - x_j\|^2}{2\delta^2}\right), \delta > 0,
$$
(7)

where $\delta$ is the hyperparameter.

### 2.3. Wavelet Packet Decomposition

Wavelet packet decomposition combines wavelet transform and multi-resolution approximation. More detailed features are extracted as the signal is subdivided step-by-step [28]. After performing a wavelet packet decomposition of level N, $2^N$ different waveform signals $D_{Nj}$, $(j = 1, 2, \ldots, 2^N - 1)$ with low to high frequencies are generated. The energy of each band signal can be calculated by Equation (8).

$$
E_{Nj} = \int |D_{Nj}(t)|^2 dt = \sum_{k=1}^{n} |d_{jk}|^2,
$$
(8)

where $d_{jk}$ is the amplitude of the $k$ point of the reconstructed signal $D_{Nj}$, and $n$ is the number of discrete points. The energy of each frequency band can be calculated from Equation (8), and the feature vector $S_N$ can be constructed from these energy values as in Equation (9).

$$
S_N = [E_{N0}, E_{N1}, \ldots, E_{Nj}, \ldots, E_{N(2^N-1)}]
$$
(9)

$S_N$ can be normalized by the min-max normalization method to obtain the new feature vector $S_N'$.

$$
S_N' = [E_{N0}', E_{N1}', \ldots, E_{Nj}', \ldots, E_{N(2^N-1)}']
$$
(10)

Figure 2 shows an example of decomposition with 4-layer wavelet packets from the experimental data in this work.

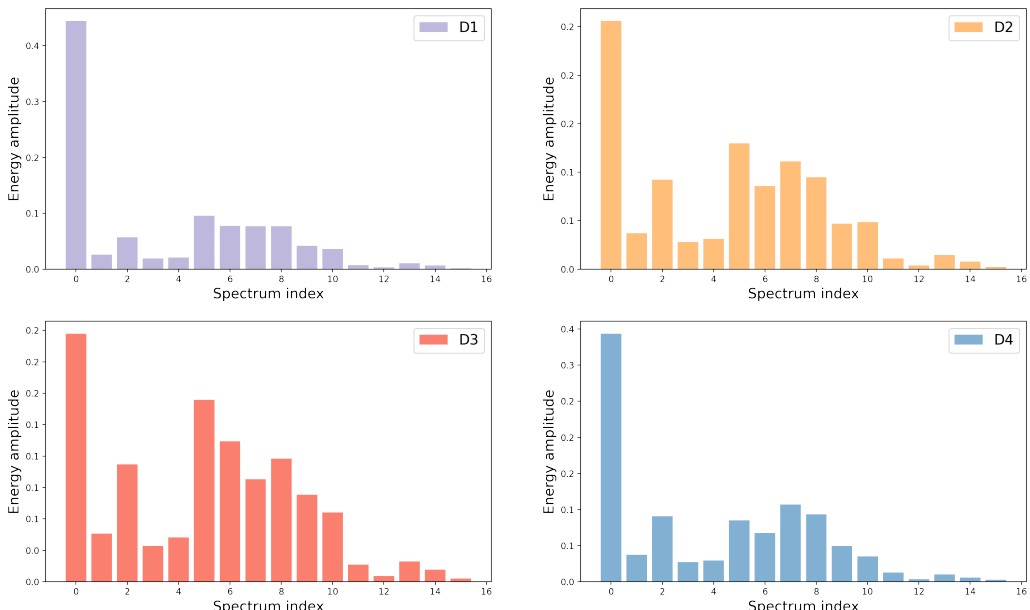

**Figure 2.** The feature vectors obtained by wavelet packet decomposition under the different damage cases. The 4 damage case are represented by D1–D4.

## 3. Experiments

### 3.1. Dataset

In this paper, the vibration response signals were collected through an eight-layer steel frame structure; each layer was 35 cm long and 25 cm wide, with a height of 20 cm between the two layers. Anchor bolts were used to fasten the bottom of the frame to the ground, while double-row bolts were used to join the beam to the columns. The diagram of the frame is shown in Figure 3. White noise excitation was applied at the third layer of the frame, and eight acceleration sensors were installed at each level along the direction of external excitation to record the acceleration response. The white noise generator model was RIGOL DG-1022, an electromagnetic exciter was used as the actuator, and the type of white noise was pre-defined (Ex1-Ex10). The model structure's steel material had an elasticity modulus of $E = 2.0 \times 10^{11}$ Pa and a density of $\rho = 7.8 \times 10^3$ kg/m$^3$. Each column member was made of a $200 \times 30 \times 3$ mm steel plate in a undamaged condition. The damage to the steel frame structure was achieved by reducing the stiffness of the steel plates (i.e., replacing the current plate with thinner ones: $200 \times 30 \times 2.5$ mm). In the vibration experiments, the duration of each recording was 32 s, the sampling frequency was 128 Hz, and a record contained 4096 data points for one sensor. Ten damage states were set up for the experiments, and the damage locations and severity for the ten damage cases are shown in Table 1.

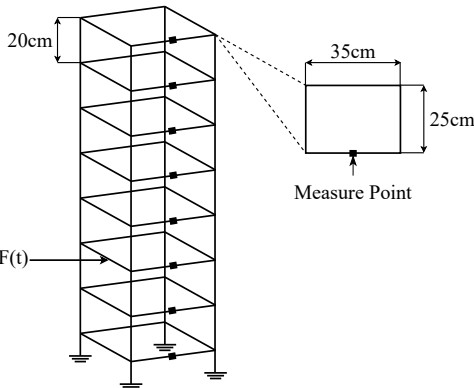

**Figure 3.** Eight-layer steel frame diagram.

**Table 1.** Description of different damage cases

| Case | Location | Decreased Stiffness (%) |
|------|----------|-------------------------|
| UD | - | 0 |
| D1 | 3 | 8.3 |
| D2 | 3 | 16.7 |
| D3 | 5 | 8.3 |
| D4 | 5 | 16.7 |
| D5 | 7 | 8.3 |
| D6 | 7 | 16.7 |
| D7 | 3 & 5 | 8.3 (both layers) |
| D8 | 3 & 7 | 8.3 (both layers) |
| D9 | 5 & 7 | 8.3 (both layers) |

In Table 1, UD is the undamaged state, D1–D6 are the cases of single-layer damage, and D7-D9 are the cases of two-layer damage. Ten different white noise excitations (Ex1-Ex10) were applied for each damage case. For each noise effect, one record was gathered. A total of 100 data were collected under 10 kinds of damage and 10 kinds of noise. Each record contains the vibration response of eight sensors, with a sampling time of 32 s and a sampling frequency of 128 Hz, for a total of 32,768 data points. Figure 4 shows the structural vibration response for the UD and D1 cases.

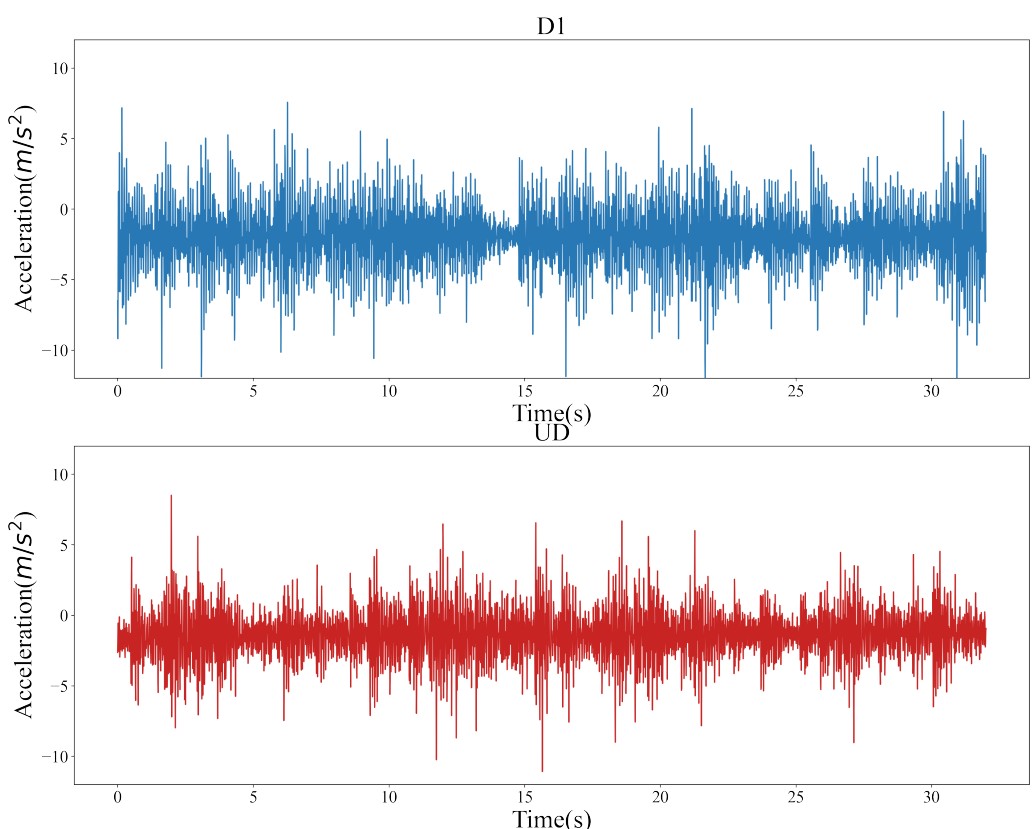

**Figure 4.** Examples of two kinds of damage data.

### 3.2. Data Preprocessing

A data preprocessing operation was needed to change the original data into a more suitable form that meets the requirements of the model. In this work, preprocessing contains four parts: (1) eliminate offset; (2) min–max normalization; (3) data slicing; and (4) splitting the training and validation sets.

### 3.2.1. Offset Elimination

During vibration testing, sensors or acquisition devices are likely to offset due to their performance problems or environmental disturbances (e.g., temperature, power supply, etc.). The offset will directly affect the accuracy of signal analysis and should be eliminated. In this paper, the elimination of offset was achieved by subtracting the mean value from the samples. For the time series $X = \{x_1, x_2, \ldots, x_n\}$, the formula for eliminating the offset is as follows:

$$\hat{X} = X - \frac{1}{n} \sum_{i=1}^{n} x_i \tag{11}$$

### 3.2.2. Data Normalization

From Figure 4, it can be found that there are differences in the vibration magnitudes of different samples. To eliminate the differences in the magnitudes of different samples and improve the classification performance [29], a min–max normalization method [30] expressed by Equation (12) was used to normalize all sample magnitudes to the range of 0 to 1.

$$\hat{X} = \frac{X - \min(X)}{\max(X) - \min(X)} \tag{12}$$

### 3.2.3. Data Slicing

In order to make full use of the data, this paper adopts a slicing approach to enhance the data information. For example, one original time series contained vibration responses recorded by eight sensors, and each vibration response contained 4096 data points. The slice length $Ns$ and the sliding step $s$ were chosen to divide the sample into multiple slices, and each slice had the same class label as the original time series. Thus, by varying $Ns$ and $s$, different numbers of training samples can be obtained, and this method is particularly effective in the case of insufficient data samples. In this work, we set Ns = 1024, s = 100, and the data was expanded from 100 to 3000 using data slicing. Figure 5 shows the detailed method of data slicing.

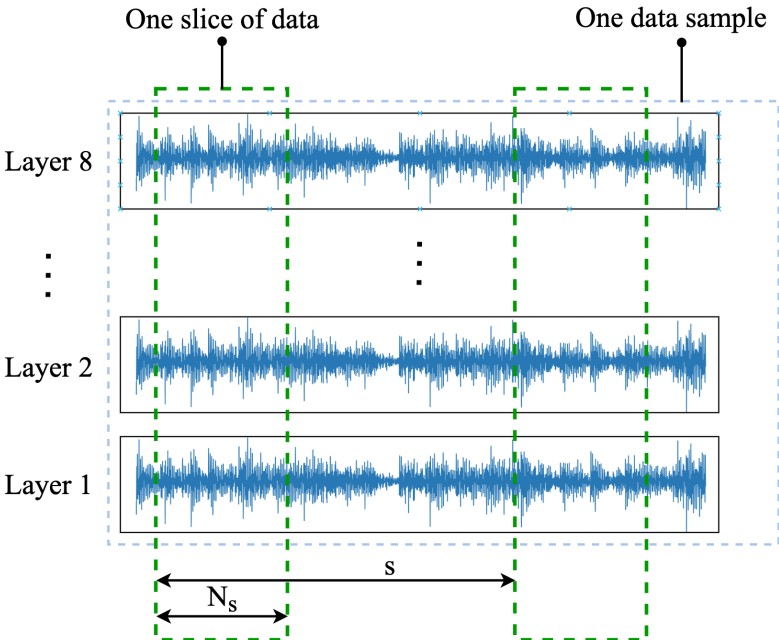

**Figure 5.** The illustration of data slicing.

### 3.2.4. Dataset Splitting

To validate the performance of the proposed model, the dataset is usually partitioned into training and test sets. During the model's training, only data from the training set is used to train it, and when testing the model's performance, the model is cross-validated using data from the test set that the model has never seen before. The process of $k$-fold cross-validation is one of the widespread cross-validation methods. The original sample is randomly partitioned into $k$ equal sized subsamples in $k$-fold cross-validation, a single subsample from the $k$ subsamples is retained as test data for evaluating model effectiveness, and the remaining $k - 1$ subsamples are used as training data. The cross-validation process is then repeated $k$ times, with each of the $k$ subsamples supplied as test data exactly once. In this paper, $k$ was set to five.

### 3.3. Baselines

We compared our model with the following baseline models:

- SVM: The feature vector was obtained by four-layer wavelet packet decomposition, and then SVM was used to identify both the damage location and the damage severity.
- 1D-CNN: Using 1D-CNN to identify both damage location and damage severity, the structure of 1D-CNN was the same as the 1D-CNN used in the method proposed in this paper.
- 1D-CNN and1D-CNN: After identifying the damage location using a 1D-CNN, the damage severity was identified using another 1D-CNN. The structure of the two 1D-CNNs were consistent with the 1D-CNN in the method proposed in this paper.

### 3.4. CNN Configurations

The structure and details of the specific parameters of the 1D-CNN used in this paper are shown in Table 2.

**Table 2.** Configuration of the 1D-CNN used in this paper.

| Layer | Output Shape | Parameter | Activation | Variables |
|---|---|---|---|---|
| Input | $1024 \times 8$ | None | None | 0 |
| Convolution 1-D | $1021 \times 8$ | Kernel number: 4; Kernel size: $8 \times 8$; | ReLU | 264 |
| Convolution 1-D | $1014 \times 16$ | Kernel number: 8; Kernel size: $16 \times 8$; | ReLU | 1040 |
| Max Pooling 1-D | $507 \times 16$ | Kernel number: 2; | None | 0 |
| Convolution 1-D | $500 \times 16$ | Kernel number: 8; Kernel size: $16 \times 8$; | ReLU | 2064 |
| Global Average Pooling 1-D | 16 | None | None | 0 |
| Dropout | 16 | None | None | 0 |
| Dense | 7 | None | Softmax | 119 |
| Total parameters | | | | 3487 |

### 3.5. Experimental Results

The proposed model was trained on a server equipped with an Intel Xeon Silver 420 (10) @ 2.194GHz CPU and an Nvidia Tesla V100 (32GB) GPU. The model was developed using the Python (version 3.7.13) programming language with the Python modules Keras (version 2.2.4) and Pycaret (version 2.3.10). We used Keras to build the 1D-CNN model and Pycaret to build the SVM model.

We adopted five-fold cross-validation to train and validate the models. The experimental results of the proposed method compared with other baseline methods are given in Table 3. From Table 3, we can see that the model performance of only using SVM combined with four-layer wavelet packet decomposition was the worst, with an average accuracy of 75.7%. This might be because the complexity of the problem to detect the damage locations and damage severity simultaneously exceeds/ed the learning ability of the model, and the informative features for classification were not well extracted. We can also find that the

method using 1D-CNN worked better than the method using only SVM, which verifies the powerful feature extraction ability of 1D-CNN in this task.

Moreover, the two-stage approach 1D-CNN and 1D-CNN worked better than the other directly identification approach, which verifies the effectiveness of the idea of "divide-and-conquer". In addition, the method using 1D-CNN and SVM achieved the best results, slightly higher than that of 1D-CNN and 1D-CNN. The specific experimental results of the two-stage methods are shown in Table 4. Since the 1D-CNN was used to identify the damage locations in the first stage in both two methods, their accuracies of detecting damage locations were very close, while in the second stage, the SVM was able to maintain a stable accuracy of 100%, which is better than that of 1D-CNN, indicating that SVM can achieve better results on the case of the small number of samples. In addition, deep learning-based methods need sufficient training samples; otherwise, they fall into an overfitting state and lower the generalization performance of the model.

**Table 3.** The 5-fold cross-validation results of the 4 methods on the test set.

|  | **SVM** | **1D-CNN** | **1D-CNN&1D-CNN** | **1D-CNN&SVM** |
|---|---|---|---|---|
| Fold 1 | 0.75 | 0.9718 | 0.9833 | 0.9966 |
| Fold 2 | 0.6964 | 0.9364 | 0.9921 | 1.0 |
| Fold 3 | 0.7143 | 0.9833 | 0.9845 | 0.9983 |
| Fold 4 | 0.8036 | 0.9718 | 0.9743 | 1.0 |
| Fold 5 | 0.8214 | 0.9645 | 0.9874 | 0.9989 |

**Table 4.** Comparison of 1D-CNN and 1D-CNN with 1D-CNN and SVM in two-stage classification performance.

|  | **1D-CNN&1D-CNN** | | **1D-CNN&SVM** | |
|---|---|---|---|---|
|  | **Location** | **Severity** | **Location** | **Severity** |
| Fold 1 | 0.9989 | 0.9743 | 0.9984 | 1.0 |
| Fold 2 | 1.0 | 0.9734 | 1.0 | 1.0 |
| Fold 3 | 0.9968 | 0.9876 | 0.9991 | 1.0 |
| Fold 4 | 0.9937 | 0.9804 | 1.0 | 1.0 |
| Fold 5 | 1.0 | 0.9856 | 0.9994 | 1.0 |

Figure 6a,b show the confusion matrix of the 1D-CNN and 1D-CNN&SVM methods. It can be seen that when only 1D-CNN was used to identify the damage location and damage severity, the error was mainly concentrated on the case of different damage severity at the same damage location, such as three samples of D1 were identified as D2 and eight samples of D5 were identified as D4. The remaining six D9 samples were incorrectly identified as D6 (note that D9 represents the existence of damage in the fifth and seventh layers, and D6 represents the existence of damage in the seventh layer only), and the model only identified one of the damage locations. In contrast, in the case of using 1D-CNN and SVM, there were no erroneous samples within the confusion matrix, and the two-stage method based on 1D-CNN and SVM improved the performance of the model.

### 3.6. Further Comparison and Results Visualization

In this work, the training and testing speeds of each method were evaluated under the environment as mentioned in Section 3.5 and shown in Table 5. From Table 5, we can see that training the SVM model took much less time than that of the 1D-CNN model because training 1D-CNN requires more parameters and epochs. The time required for testing the SVM model and testing 1D-CNN model were both small, but the SVM model was still 33% faster than 1D-CNN. The training time required for the two-stage model 1D-CNN and SVM was 38.5 s, which was 10% faster than that of 1D-CNN and 1D-CNN which needed 43.2 s, and the testing time required for both methods was 1.2 s and 1.5 s, respectively.

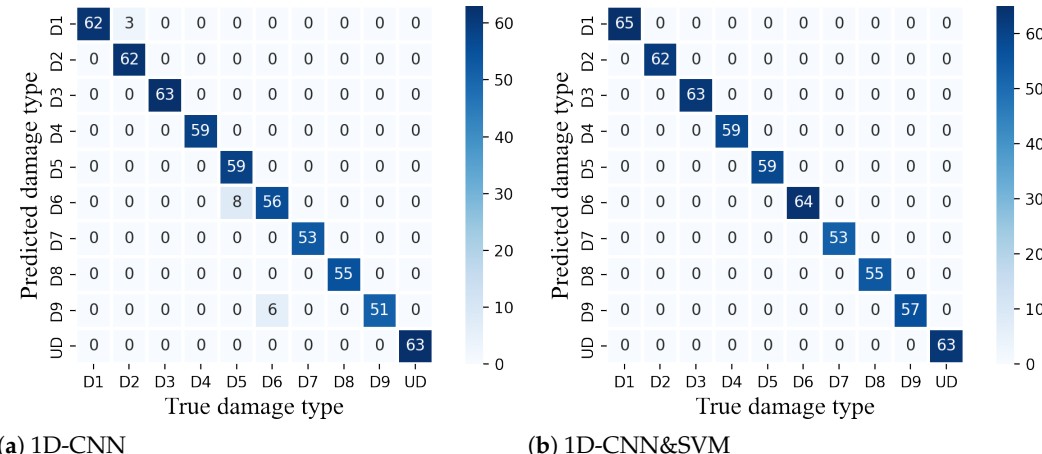

(**a**) 1D-CNN          (**b**) 1D-CNN&SVM

**Figure 6.** Confusion matrixes of 1D-CNN and 1D-CNN and SVM methods.

**Table 5.** The time required for training and testing of the 4 methods.

|       | SVM   | 1D-CNN | 1D-CNN&1D-CNN | 1D-CNN&SVM |
|-------|-------|--------|---------------|------------|
| Train | 9.1 s | 31.5 s | 43.2 s        | 38.5 s     |
| Test  | 0.6 s | 0.9 s  | 1.5 s         | 1.2 s      |

We further adopted t-distributed stochastic neighbor embedding (T-SNE) to visualize the classification results with the proposed model. T-SNE is a nonlinear dimensionality reduction method that can reduce high-dimensional data to two or three dimensions for visualization [31]. The T-SNE results of the test samples before and after classification are shown in Figure 7a,b, respectively. Different colors represent different types of damage, and it can be found that the sample distribution was chaotic and nearly indistinguishable before classification; however, after classification, samples of different damage types were clustered together separately, indicating that the method proposed in this paper has a strong classification ability for structural vibration response samples.

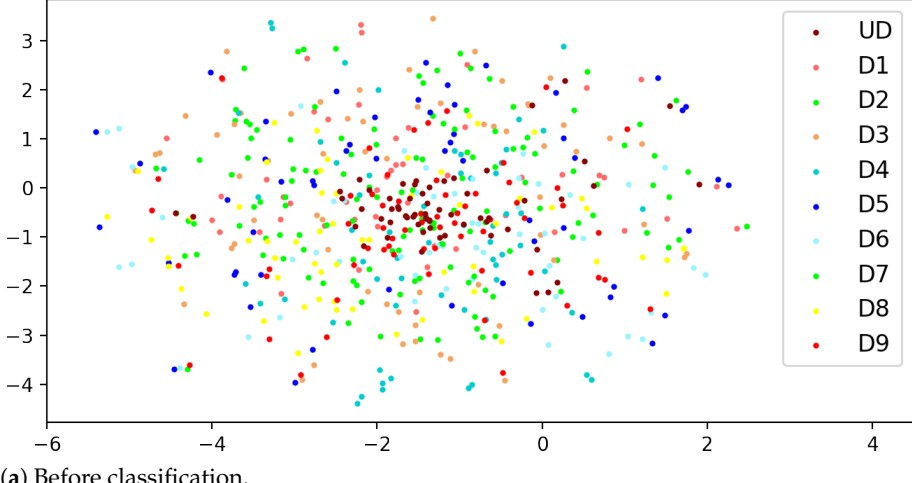

(**a**) Before classification.

**Figure 7.** *Cont.*

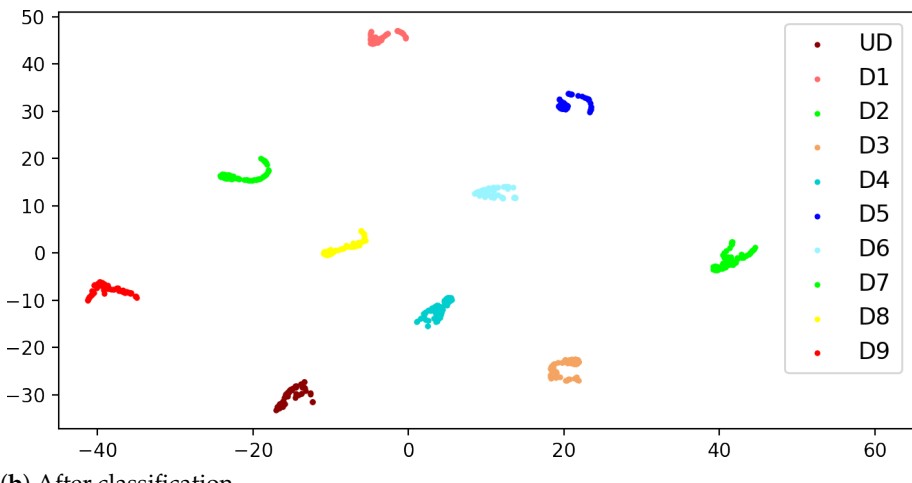

(**b**) After classification.

**Figure 7.** Visualization results of the test samples before and after classification.

## 4. Discussion

The health of buildings is significant for the safety of human life. Damage detection by structural damage response is a relatively popular method in SHM. With the development of machine learning and deep learning, more and more methods are being applied to this field. Machine learning requires fewer training samples and low computational cost but requires manually designed complex feature extraction methods. Deep learning can automatically extract feature, but has a high computational cost and requires more training samples.

We propose a two-stage structural damage response method based on 1D-CNN and SVM by analyzing the vibration damage response of an eight-layer steel framework. The 1D-CNN is used in the first stage to detect the damage location, and the SVM is used in the second stage to detect the damage severity, which fully combines the advantages of 1D-CNN and SVM to achieve better damage detection with less computational cost and simpler feature extraction methods and is meaningful for the application of SHM.

In this paper, we compare our proposed method with several other methods. From Table 3, it can be found that, in general, the two-stage damage detection methods work better than the single-stage damage detection method. Among the single-stage damage detection methods, 1D-CNN is much better than SVM. On the one hand, it is because 1D-CNN has an extremely strong feature extraction ability, and on the other hand, it is because we have not designed a feature extraction method for SVM that combines expertise, but directly uses wavelet packet decomposition as the feature extraction method, resulting in a large difference between the two effects. In Lei et al. [11], the authors achieved extremely high accuracy by using variance, regression coefficients, and cross-correlation function amplitude as features, followed by SVM for classification. This suggests that better results can be achieved with SVM if the designed feature extraction method is good enough, but this requires strong expertise, and the designed feature extraction method may not be applicable to other structures. In contrast, 1D-CNN only uses a simpler structure, which is good at automatically extracting features and achieving a high accuracy rate and is applicable to a variety of structures.

Among the two-stage methods, the proposed method in this paper is slightly better than 1D-CNN and 1D-CNN. It can be seen from Table 4 that the two methods are close to each other in detecting the damage location because the first stage of both methods is same. The main difference is in the second stage of damage severity detection. To further probe the reason, we compared the confusion matrix of 1D-CNN with 1D-CNN and SVM. From Figure 6a, we can find that the samples incorrectly identified by 1D-CNN were all samples with the same damage location and different damage severity. On the one hand, this is because the change of damage response caused by the change of damage severity was small, and on the other hand, it is because the number of samples with different damage

severity at the same damage location was small, which was not enough for 1D-CNN to learn sufficient information. While SVM uses the wavelet packet decomposition of the damage response as the feature vector, which is a classification problem with high-dimensional small samples, and is well suited to be solved by using SVM, 1D-CNN and SVM have indeed achieved better results.

There are not many studies on multi-level damage detection. Shao et al. [32] proposed a multilevel damage classification method based on Lamb wave and transfer learning. They divided the damage detection into three levels, which detected the existence, location, and size of damage. In future work, we will consider adding a stage to detect the existence of damage. Their method used 1D-CNN for damage detection at all three levels, and although their method achieved high accuracy, it also takes more time to train. They also realized this problem, so they used the transfer learning method to share part of the structure and weights of the 1D-CNN in all three levels, which makes the training faster and saves more time. If they consider using SVM to detect the size of the damage, they may be able to save more time while ensuring the accuracy is not reduced.

Our study also has many limitations. In real-world applications, the location and severity of damage are continuous, whereas the experiments in this paper have only limited classes of damage locations and severity and use a classification foundation model rather than a regression model, which makes the method in this paper unable to predict the type of damage outside the dataset.

In future work, we intend to replace the classification model by employing a regression model and designing more types of damage locations and damage severity so that the model can accurately discriminate between types of damage outside the dataset.

## 5. Conclusions

In this paper, a two-stage structural damage detection method based on 1D-CNN and SVM is proposed. It is still challenging to detect the damage of a structure accurately based on the vibration response of the structure. To solve the problem that traditional machine learning methods need to design feature extraction methods by manually combining expert knowledge, this paper uses 1D-CNN to automatically extract rich features from vibration responses. To solve the problem that the number of samples with different severity of damage at the same damage location is small and 1D-CNN cannot distinguish these samples well, this paper uses SVM combined with wavelet packet decomposition to achieve the accurate identification of these samples. Experiments were conducted on an eight-layer steel frame, and damage responses were collected for ten damage cases. After preprocessing operations such as offset elimination, normalization, and slicing, the damage locations corresponding to the samples were first determined by 1D-CNN, and then the damage severity corresponding to the samples was determined by SVM. In the comparison experiments with other methods, the method proposed in this paper achieved the best results, taking into account the operation speed and recognition effect.

However, the method in this paper still has some limitations, as it can only determine predefined damage cases and cannot determine continuous damage locations or damage severity. In future research, we intend to use deep learning-based regression methods to predict continuous damage and combine expert knowledge to improve detection performance. In addition, the data are provided by other labs, and we do not have permission to share the data. The code is released at https://github.com/jch12138/two-stage-structure-damage-detection (accessed on 30 August 2022).

**Author Contributions:** Data curation, J.L. and X.W.; methodology, C.J. and Q.Z.; software, C.J.; supervision, Q.Z. and J.L.; visualization, X.W.; writing—original draft, C.J.; writing—review & editing, Q.Z. and J.L. All authors have read and agreed to the published version of the manuscript.

**Funding:** This work is partially supported by China Natural Science Foundation under grant No. 62171391. Shaorong Fang and Tianfu Wu from Information and Network Center of Xiamen University are acknowledged for the help with the GPU computing.

**Institutional Review Board Statement:** Not applicable, this research not involving humans or animals.

**Informed Consent Statement:** Not applicable.

**Data Availability Statement:** Not applicable.

**Conflicts of Interest:** The author declares no conflict of interest.

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
