# Peer review of "A Two-Stage Structural Damage Detection Method Based on 1D-CNN and SVM"

_applsci, doi:10.3390/app122010394_

Round 1

Reviewer 1 Report

This paper presents a two-stage structural damage detection method based on 1D convolutional neural networks and support vector machine. The damage location and severity are detected by the method. The following comments should be addressed before the acceptance of the paper.

What does SVM stand for? It is not clear abridged in the draft.

The meanings of h and u in equation (1) are unclear. What do they represent?

More explainations are needed for the activation function in (2).

More statements should be given to the experimental setup. How do you connect the plate to the frame?

Can the current two-stage sheme be used for identification of damages of other values of declined stiffness in Table 1? For example, if the declined stiffness is increased, can you still detect the updated damage, by only using the existing data and training model in the current paper?

How to use current scheme to identify the damage with its location different from those in Table 1? Do we need a new training again?

Author Response

Response to the Reviewers

Dear Editor and Reviewers,

Thank you very much for your helpful comments and suggestions. We have carefully gone through the review comments and have made revisions based on the reviewer comments. We fist summarize our revisions as follows.

  • We have corrected grammatical errors in the article and added explanations to some equations.
  • We have added more content in the conclusion section. (See Section 5)
  • We have added discussion section. (See Section 4)

The detailed response to the individual reviewers is listed below.

To Reviewer #1

Comment 1: What does SVM stand for? it is not clear abridged in the draft.

Response: Thanks for your comment. SVM stands for Support Vector Machine, we have added the introduction of the abbreviation for SVM in Section 1.

Comment 2: The meanings of h and u in equation (1) are unclear. What do they represent?

Response: Thanks for your comment. In equation (1), h is the subsequence, u is the kernel. The introduction of h, u is added below equation (1).

Comment 3: More explanations are needed for the activation function in (2).

Response: Thanks for your comment. We have added more explanations for equation (2) .

Comment 4: More statements should be given to the experimental setup. How do you connect the plate to the frame?

Response: Thanks for your comment. Anchor bolts are used to fasten the bottom of the frame to the ground, while double-row bolts are used to join the beam to the columns. This has been added in Section 3.1.

Comment 5: Can the current two-stage scheme be used for identification of damages of other values of declined stiffness in Table 1? For example, if the declined stiffness is increased, can you still detect the updated damage, by only using the existing data and training model in the current paper?

Response: Thanks for your comment. We only have samples with two levels of damage severity in this dataset. The model can be retrained using new kinds of damage data and identify this kind of damage. But the model cannot identify the untrained damage case since it an offline model.

Comment 6: How to use current scheme to identify the damage with its location different from those in Table 1? Do we need a new training again?

Response: Thanks for your comment. I am sorry that our scheme cannot be applied directly to locations different from Table 1. In order to detect more damage locations, the dimensionality of the model output vector needs to be increased, so the model needs to be retrained.

Reviewer 2 Report

The presented paper deals with an important subject of machine-assisted structural damage detection. The proposed concept of combining Deep learning (1D-CNN) for damage localization and Machine learning (SVM) for severity detection is interesting and revealed to be efficient.

However, some mathematical notations and Lemma presentations are not rigorous enough to correctly understand the contents of the paper. The authors are requested to recheck all the definitions of variables and further clarify these equations. The experimental setup, some results, and figures are not commented on in detail. In the reviewer’s opinion, the paper should be revised before being accepted for publication. Please, find below the full list of remarks.

Line 37: the SVM abbreviation is not introduced

Line 51: the CNN abbreviation is not introduced (it is introduced in the abstract only)

Equations (1) and (2): variables x, y, h, N, and others are not introduced

Line 119 and below equation (4): the name of Lagrange must be capitalized 

The relationship between equations (9) and (10) is unclear. How is the normalization performed? 

Equation (3) is not clearly introduced. For example, the meaning of the m variable is not explained, φ(xi) does not appear in the equation. Please, explain the meaning of the equation in more detail. 

Equations (4) – (7): presentation is not very clear; Φ and δ are not introduced

Equations (8): what are the integration limits? 

Figure 2 caption: legend entries D1 – D4 must be introduced

Line 129: Could the authors present in the paper which equipment was used to introduce white noise vibrations (white noise generator + type of actuator)? If the noise signals were completely random (uncorrelated noise), or some set of prerecorded noise sequences were used? 

Table 1: possibly, it is better to use the term ‘Decreased stiffness’ instead of ‘Declined stiffness’. Please use (both layer) -> (both layers).

Line 133: For ease of reading, could the authors indicate the number of points in each record?

Line 141: repetition of line 133.

Figure 4. Supposing the used noise generator provides uncorrelated sequences. In that case, the presentation of two white noise realizations may not be reasonable as it depends more on the external noise source than the structural damage. This remark is related to the comment for Line 129. 

Figure 5: It might be unnecessary as it illustrates some very basic concepts of offset elimination. 

Lines 170 and 171. K variable is used simultaneously in capitalized (K) and not capitalized (k) format. Please, use the italic font for the variables.  

Line 230: The T-SNE abbreviation is not introduced.

Figure 8. Could the authors please explain this figure in more detail? Which units are represented by the figure axis?

Discussion on data and code availability could be included in the conclusion section.

To have an unbiased view in the paper, there should be some discussions on the limitations of the proposed method in the conclusion section. 

Author Response

Response to the Reviewers

Dear Editor and Reviewers,

Thank you very much for your helpful comments and suggestions. We have carefully gone through the review comments and have made revisions based on the reviewer comments. We fist summarize our revisions as follows.

  • We have corrected grammatical errors in the article and added explanations to some equations.
  • We have added more content in the conclusion section. (See Section 5)
  • We have added discussion section. (See Section 4)

The detailed response to the individual reviewers is listed below.

To Reviewer #2

Comment 1: Line 37: the SVM abbreviation is not introduced.

Response: Thanks for your comment. SVM stands for Support Vector Machine, we have added the introduction of the abbreviation for SVM in Section 1.

Comment 2: Line 51: the CNN abbreviation is not introduced (it is introduced in the abstract only).

Response: Thanks for your comment. CNN stands for Convolutional Neural Network, we have added the introduction of the abbreviation for CNN in Section 1.

Comment 3: Equations (1) and (2): variables x ,y, h, N, and others are not introduced.

Response: Thanks for your comment. We have added a description of equations (1) and (2). In equation (1), y is the output signal, N is the length of the kernel, h is the subsequence. In equation (2), x is the input of ReLU. The introduction of x, y, h, and N is added below equation (1).

Comment 4: Line 119 and below equation (4): the name of Lagrange must be capitalized.

Response: Thanks for your comment. We have changed the two occurrences of ‘Lagrange’ below equation (3) and equation (4) to upper case.

Comment 5: The relationship between equations (9) and (10) is unclear. How is the normalization performed?

Response: Thanks for your comment. We use the min-max normalization method, which scales the range of features to 0-1. It is mentioned in Section 3.2.2.

Comment 6: Equation (3) is not clearly introduced. For example, the meaning of the m variables is not explained,  does not appear in the equation. Please explain the meaning of the equation in more detail.

Response: Thanks for your comment. We added more clarification to equation (3) by adding the interpretation of the variable m.  is redundant here and we have removed it.

Comment 7: Equations (4)-(7): presentation is not very clear;  and  are not introduced.

Response: Thanks for your comment. We added introduction of .  is redundant here and we have removed it.

Comment 8: What are the integration limits?

Response: Thanks for your comment. The integration limits in equation (8) are the entire time axis, form negative infinity to positive infinity.

Comment 9: Figure 2 caption: legend entries D1-D4 must be introduced.

Response: Thanks for your comment. We have introduced D1-D4 in the title of Figure 2.

Comment 10: Line 129: Could the authors present in the paper which equipment was used to introduce white noise vibrations (white noise generator + type of actuator)? If the noise signals were completely random (uncorrelated noise), or some set of prerecorded noise sequences were used?

Response: Thanks for your comment. In this paper, the white noise generator model is RIGOL DG-1022. An electromagnetic exciter is used as the actuator. The type of white noise is pre-defined (Ex1-Ex10). We have added these to the article in Section 3.1.

Comment 11: Table 1: possibly, it is better to use the term ‘Decreased stiffness’ instead of ‘Declined stiffness’. Please use (both layer) -> (both layers).

Response: Thanks for your comment. 'Decreased stiffness' is more accurate, and I apologize for not noticing the mistake of 'both layer'. I have corrected it as you said. Thank you!

Comment 12: Line 133: For ease of reading, could the authors indicate the number. Of points in each record?

Response: Thanks for your comment. We have indicated the number of points in each record in Section 3.1.

Comment 13: Line 141: repetition of line 133.

Response: Thanks for your comment. We have indicated the number of points in each record in Section 3.1.

Comment 14: Figure 4. Supposing the used noise generator provides uncorrelated sequences. In that case, the presentation of two white noise realizations may not be reasonable as it depends more on the external noise source than the structural damage. This remark is related to the comment for Line 129.

Response: Thanks for your comment. In fact, the two records in Figure 4 correspond to the structural responses of D1 and UD under the same external excitation (white noise type). So the difference in the waveform features of both can perhaps be explained more by the structural damage.

Comment 15: Figure 5. It might be unnecessary as it illustrates some very basic concepts of offset elimination.

Response: Thanks for your comment. Figure 5 has been removed.

Comment 16: Lines 170 and 171. K variable is used simultaneously in capitalized (K) and not capitalized (k) format. Please, use the italic font for the variables.

Response: Thanks for your comment. We have changed the k in italic font in Section 3.2.4.

Comment 17: Line 230. The T-SNE abbreviation is not introduced.

Response: Thanks for your comment. T-SNE stands for T-distributed Stochastic Neighbor Embedding. We have added the introduction of the abbreviation for T-SNE in Section 3.6.

Comment 18: Figure 8. Could the authors please explain this figure in more detail? Which units are represented by the figure axis?

Response: Thanks for your comment. T-SNE is a technique for visualizing high-dimensional data in a low-dimensional space. The axes are not meant to be interpretable, they just define a 2D space into which higher dimensional space will be projected, preserving relative proportional distances as much as possible.

Comment 19: Discussion on data and code availability could be included in the conclusion section.

Response: Thanks for your comment. We can provide the code, but the data is provided by other labs and we are not authorized to disclose the data, so unfortunately the data used in this paper is not available. We have added this note in the conclusion section.

Comment 20: To have an unbiased view in the paper, there should be some discussions on the limitations of the proposed method in the conclusion section.

Response: Thanks for your comment. We have added the limitations of the method proposed in this paper in the conclusion section.

Reviewer 3 Report

It's a good topic. The study will contribute to the literature. I have a few suggestions to make the study more effective.

1)In the abstract section, numerical information showing the success of the proposed approaches should be presented.

2)What is the innovative aspect of this article? In the light of the literature, please give in the Introduction section.

3)Why did you choose to propose a CNN and SVM-based approach?

4)This study does not have a Discussion section. Please add a Discussion section.

5)The conclusion section is very short. The conclusion section should be expanded to best describe the study.

6)I suggest updating the literature with studies in 2020 and beyond.

7)How does this work contribute to future work?

Author Response

Response to the Reviewers

Dear Editor and Reviewers,

Thank you very much for your helpful comments and suggestions. We have carefully gone through the review comments and have made revisions based on the reviewer comments. We fist summarize our revisions as follows.

  • We have corrected grammatical errors in the article and added explanations to some equations.
  • We have added more content in the conclusion section. (See Section 5)
  • We have added discussion section. (See Section 4)

The detailed response to the individual reviewers is listed below.

To Reviewer #3

Comment 1: In the abstract section, numerical information showing the success of the proposed approaches should be presented.

Response: Thanks for your comment. We have added our experimental results to the abstract section.

Comment 2: What is the innovative aspect of this article? In the light of the literature, please give in the introduction section.

Response: Thanks for your comment. We have listed the main contributions of this paper at the end of the introduction. The main contributions of our work are as follows:

  • We propose a new two-stage structural damage detection method which follows the strategy of "divide-and-conquer" to solve the problem of insufficient training data and enhance the model performance for multi-level structural damage detection.

  • We verify the proposed model on an eight-level steel frame structure, the experimental results show that the proposed method outperforms the state-of-the-art methods in terms of both damage location detection and damage severity detection.

Comment 3: Why did you choose to propose a CNN and SVM-based approach?

Response: Thanks for your comment. 1D-CNN has achieved amazing results in the classification of time series; therefore, we choose 1D-CNN for damage detection. However, 1D-CNN-based model requires more data for training, and the dataset for structural damage detection is usually small, thus directly using 1D-CNN to identify the location and severity of damage is likely to be underfitted and cannot achieve better results. SVM is a kind of strong learning machine especially for small-sized training dataset problem and it has achieved widely uses in many data mining fields. Therefore, in the case of the damage location is determined, detecting the severity of damage becomes a small sample binary classification problem, which is very suitable for using SVM to solve the problem.

Comment 4: This study does not have a Discussion section. Please add a Discussion section.

Response: Thanks for your comment. We have added discussion section as Section 4.

Comment 5: The conclusion section is very short. The conclusion section should be expanded to best describe the study.

Response: Thanks for your comment. We have added more content in the conclusion section.

Comment 6: I suggest updating the literature with studies in 2020 and beyond.

Response: Thanks for your comment. We have replaced the older studies with two newer studies in the introduction section.

Comment 7: How does this work contribute to future work?

Response: Thanks for your comment. Our model currently achieves good results for the classification task, and future work is directed towards improving the classification model to a regression model that achieves more accurate damage detection and can detect the location and severity of damage outside the dataset.

Round 2

Reviewer 1 Report

More descriptions on the details of the setup of the frame system are recommended to be added.

Reviewer 3 Report

The following recommendations have not been sufficiently taken into account. So I repeat my criticisms:

-What is the novelty of the study? So far, the novelty of the study has not been revealed.

-The discussion section is short and inadequate. An effective discussion is required.

-Conclusion section is still short. In the conclusion part, a general framework of the study should be drawn.

Round 3

Reviewer 3 Report

The manuscript in the system is the same as before. The changes specified by the author are not included  here. It may have been submitted incorrectly. It needs to be checked.